# *Tinamiphilopsis temmincki* sp. n., a New Quill Mite Species from Tataupa Tinamou, and the Early History of Syringophilid Mites [note 1]

**DOI:** 10.3390/ani13172728

**Published:** 2023-08-28

**Authors:** Maciej Skoracki, Monika Fajfer, Martin Hromada, Jan Hušek, Bozena Sikora

**Affiliations:** 1Department of Animal Morphology, Faculty of Biology, Adam Mickiewicz University, Uniwersytetu Poznańskiego 6, 61-614 Poznań, Poland; 2Department of Molecular Biology and Genetics, Institute of Biological Sciences, Cardinal Stefan Wyszynski University, Wóycickiego 1/3, 01-938 Warsaw, Poland; m.fajfer@uksw.edu.pl; 3Laboratory and Museum of Evolutionary Ecology, Department of Ecology, Faculty of Humanities and Natural Sciences, University of Prešov, 08001 Prešov, Slovakia; hromada.martin@gmail.com; 4National Museum of the Czech Republic, Václavské námestí 68, 11579 Prague, Czech Republic; jan.husek@nm.cz

**Keywords:** Acari, birds, ectoparasites, phylogeny, Syringophilidae, tinamous

## Abstract

**Simple Summary:**

This research presents a description of a new species of quill mite, *Tinamiphilopsis temmincki* sp. n. (Acariformes: Syringophilidae), which was found on a representative of palaeognathous bird species, the Tataupa Tinamou (*Crypturellus tataupa*), in South America. Alongside describing this new species, a phylogenetic analysis was conducted on the primitive syringophilid genera. The results indicate that the genus *Tinamiphilopsis* is nested among the syringophilid genera associated with neognathous birds. This placement has significant implications for understanding the evolutionary relationship between quill mites and their avian hosts.

**Abstract:**

The quill mite fauna of the Syringophilidae family (Acariformes: Prostigmata), which is associated with palaeognathous birds of the Tinamiformes order, remains poorly studied. Thus far, only two species of syringophilid mites have been documented on four species of tinamous. In this study, we present a description of a new species, *Tinamiphilopsis temmincki* sp. n., which was found on the Tataupa Tinamou (*Crypturellus tataupa*) in South America. This newly identified species differs from others in the genus due to the short hysteronotal setae *d2* in females, unlike the long setae *d2* found in females of other *Tinamiphilopsis* species. In addition to describing the new species, we conducted a phylogenetic analysis of the primitive syringophilid genera. The results reveal that the *Tinamiphilopsis* genus does not emerge as a sister group to all other syringophilids. Instead, it is deeply embedded within the radiation of quill mites associated with neognathous birds. This study provided evidence that mites belonging to the genus *Tinamiphilopsis* initially parasitised Neoavian birds before host switching to tinamous birds. This placement carries significant implications for our understanding of the evolution of quill mites and their relationship with their avian hosts.

## 1. Introduction

Tinamidae (Tinamous), the only family in the order Tinamiformes, consists of small- to medium-sized birds found in Central and South America. This family comprises 47 species in nine genera and two subfamilies, Tinaminae and Nothurinae [1,2]. Birds of this family are widespread geographically and are associated with woodland and open grassland habitats from Southern Mexico to Patagonia [3,4]. Many studies have established the monophyly of Tinamidae and their connection to flightless ratites (including ostriches, emus, and their relatives). Both groups belong to palaeognaths (Palaeognathae), an early diverging group of modern birds [5,6,7,8,9,10,11,12,13]. However, there is limited research on the relationships among tinamous themselves [14,15]. The most comprehensive study was recently presented by Almeida et al. [16] and comprised the analysis of tinamous phylogenetic relationships and divergence dates, including both living and extinct species.

Prostigmatan fauna (Acariformes: Trombidiformes: Prostigmata) that is permanently associated with tinamous is represented only by members of the family Syringophilidae, whereas representatives of the other prostigmatan families, including Harpirhynchidae, Cheyletidae (Ornithocheyletini, Metacheyletiini, and Cheletosomatini), and Ereynetidae (Speleognathinae), have never been collected from any of the palaeognathous birds [17,18,19,20,21]. Currently, the family Syringophilidae associated with Tinamidae is represented by two species of the genus *Tinamiphilopsis,* which are recorded on four species of the subfamily Nothurinae, i.e., *Tinamiphilopsis elegans* Skoracki and Sikora, 2004, collected from the elegant crested tinamou *Eudromia elegans* Geoffroy Saint-Hilaire, and *Tinamiphilopsis ariconte* Skoracki et al., 2012, recorded from the red-winged tinamou *Rhynchotus rufescens* (Temminck), the white-bellied nothura *Nothura boraquira* (Spix), and the lesser nothura *Nothura minor* (Spix) [22,23]. Our knowledge encompasses only the four host species mentioned above, representing merely 9% of tinamous diversity, which vividly demonstrates the paucity of our understanding regarding syringophilid mites from this host group.

In this paper, we present the description of a new species of syringophilid mite, *Tinamiphilopsis temmincki* sp. n., collected from a representative of the subfamily Tinaminae, the tataupa tinamou, *Crypturellus tataupa* (Temminck), from South America. We also conducted a phylogenetic analysis to examine the placement of the *Tinamiphilopsis* in relation to the most primitive genera of Syringophilidae. Our findings shed new light on the evolutionary relationships of these mites and provide important insights into their biodiversity.

## 2. Materials and Methods

### 2.1. Mites Collection and Description

Mite material was collected from the dry bird skin of *Crypturellus tataupa* deposited in the ornithological collection, which is housed in the National Museum of the Czech Republic, Prague, Czechia (NMP) (Figure 1). Under laboratory conditions, the infected quill (the wing-covert quill) was dissected. Individual mites were removed and placed in Nesbitt’s liquid for 36 h at room temperature, and then, they were mounted on slides in Faure’s medium [24]. Identifications and drawings of the mite specimens were carried out using a ZEISS Axioscope light microscope (Carl-Zeiss AG, Oberkochen, Germany) equipped with differential interference contrast optics. Drawings of the new quill mite species were made with the drawing attachment (a camera lucida).

All measurements in the description are presented in micrometers. The paratypes’ measurements are indicated in brackets, appearing after the data for the holotype. The idiosomal setation adheres to Grandjean’s classification [25] as adapted for Prostigmata by Kethley [26]. The leg chaetotaxy follows the nomenclature proposed by Grandjean [27], while the morphological terminology is in accordance with Skoracki [24]. The scientific and common names of the birds are based on Clements et al. [2].

Specimen depositories and reference numbers are abbreviated as follows: AMU—Adam Mickiewicz University, Department of Animal Morphology, Poznan, Poland; ZSM—Bavarian State Collection of Zoology, Munich, Germany.

### 2.2. Phylogenetic Analysis

#### 2.2.1. Taxa Selection

Because this study aimed to recognise the phylogenetic relationship of the genus *Tinamiphilopsis*, we included in the ingroup all mite genera that possess a full complement of setae of the idiosoma and legs (plesiomorphic feature). Considering the arguments of Yeates [28] and Prendini [29] that it is preferable to include real species in a cladistic analysis rather than supra-species taxa, the genera or each species group recognised within them is represented by 1–3 species in our analysis.

Because the monophyly of the family Syringophilidae was tested with numerous outgroups and always received high support [30,31,32], only two outgroups were used in the analyses, a free-living predator *Cheyletus eruditus* (Schrank) and a quill-inhabiting predator *Cheletopsis norneri* (Poppe), both belonging to the sister family Cheyletidae.

#### 2.2.2. Cladistic Analysis

The qualitative characters from the external morphology, such as the presence/absence of a structure or the form of specific morphological features, were used in this analysis. Only adult females were analysed because males and immatures are unknown in many included taxa. In total, 29 OTUs and 49 informative characters were included in the maximum parsimony analysis (Appendix A). The data matrix was prepared using NEXUS Data Editor 0.5.0 [33] (Appendix A).

All characters were treated as unordered, and their states were polarised using an outgroup comparison. The plesiomorphic state of each character was designated as ‘0’, the apomorphic states were designated as ‘1, 2, 3’, the missing states were designated as ‘?’, and inapplicable was designated as ‘-’. Characters with multiple states were treated as polymorphic and not modified into binary characters. The characters, such as the number of tines in the proral setae (*p*’ and *p*”), the number of chambers in the peritreme branches, and the total body lengths, were divided into multiple states.

The reconstruction of phylogenetic relationships was performed using PAUP 4.0 [34]. The heuristic search option was used for the maximum parsimony analysis. The delayed transformation option favours parallelism over reversal and was applied for a posteriori optimisation of character states and tracing of character changes in lineages. Initially, all characters were unweighted, and then successive weighting was performed according to the rescaled consistency index (RC) to find a maximally consistent tree [35,36].

## 3. Results

### 3.1. Systematic

Family: Syringophilidae Lavoipierre, 1953.

Subfamily: Syringophilinae Lavoipierre, 1953.

Genus: *Tinamiphilopsis* Skoracki and Sikora, 2004.

#### 3.1.1. Description

*Tinamiphilopsis temmincki* sp. n.

Female, holotype (Figure 2 and Figure 3): The total body length is 700 (660–750 in 11 paratypes). In the gnathosoma, the stylophore is 250 (230–250) long, and the exposed portion of the stylophore is apunctate and 190 (175–190) long. The infracapitulum is punctate in the anterior part. Each medial branch of the peritremes has one longitudinal chamber and each lateral branch has five chambers. The movable cheliceral digit is 190 (185–190) long. In the idiosoma, the propodonotal shield is well sclerotised and punctate, with a concave posterior margin, and bearing bases of all propodonotal setae except *c2*. The propodonotal setae *vi*, *ve*, and *si* are smooth. The length ratio of setae *vi*:*ve*:*si* is 1:1.4–1.8:1.7–2.4. The bases of setae *c1* and *se* are situated at the same transverse level. The hysteronotal shield is well sclerotised, fused to the pygidial shield, and apunctate, and the bases of setae *d1* are situated on the lateral margin, with the anterior margin reaching the level of setae *d2*. The bases of setae *d1* are situated closer to *d2* than to *e2*. The length ratio of setae *d2*:*d1*:*e2* is 1:2:2.3–2.4. The genital plate is well sclerotised, bearing bases of setae *ag2* and *ag3* on the lateral margins. Setae *ag1* and *ag2* are subequal in length, both slightly shorter than *ag3*. The coxal fields I–IV are well sclerotised and punctate. In the legs, the solenidia are shown in Figure 3B, and there are fan-like setae of legs III and IV with nine or ten tines.

Lengths of setae: *vi* 35 (25–35), *ve* 50 (45–60), *si* 60 (60–65), *se* 200 (175–200), *c1* 205 (185–210), *c2* 185 (195–215), *d1* 105 (100–115), *d2* 50 (50–55), *e2* 120 (120–140), *f1* 40 (35–40), *f2* 215 (180–205), *h1* 30 (25–35), *h2* 285 (300–320), *ag1* 65 (55–60), *ag2* 65 (40–60), *ag3* 80 (65–75), *ps1* and *ps2* 25 (25–30), *g1* and *g2* 35 (30–35), *l’RIII* 55 (45–55), *l’RIV* 30 (35–40), *3b* 50 (40–50), *3c* 65 (50–65), *4b* 40 (35–50), *4c* 55 (45–50), *tc’III–IV* 35 (30–35), and *tc”III–IV* 55 (45–55).

Male (Figure 4): The total body length is 570 in one paratype. In the gnathosoma, the stylophore is 200 long, and an exposed portion of the stylophore is apunctate and 160 long. The infracapitulum is covered with minute punctations in the posterior part. Each medial branch of the peritreme has one chamber and each lateral branch has six chambers. In the idiosoma, the propodonotal shield is entire and punctate, rectangular in shape, and bearing bases of all propodonotal setae except *c2*. The length ratio of setae *vi*:*ve*:*si* is 1:2:5.3. The hysteronotal shield is well sclerotised, fused to the pygidial shield, and punctate laterally. Setae *d2* is 3.7 times longer than *d1* and *e2*. Setae *h2* is about 13 times longer than *f2*. The aggenital series are represented by two pairs of setae, with setae *ag1* being slightly (1.2 times) longer than *ag2*. The coxal fields I–IV are well sclerotised and punctate; the anterior margins of coxal fields III reach above the level of setae *3a*. The cuticular striations are shown in Figure 2A,B. In the legs, there are fan-like setae of legs III and IV with nine or ten tines.

Lengths of setae: *vi* 30, *ve* 60, *si* 160, *se* 210, *c1* 190, *c2* 200, *d1* 15, *d2* 55, *e2* 15, *f2* 20, *h2* 255, *ag1* 65, *ag2* 55, *l’RIII* 50, *l’RIV* 35, *3b* 50, and *3c* 70.

##### Type Material

Female holotype and paratypes: Eleven females and one male were collected from the wing-covert quill of the tataupa tinamou, *Crypturellus tataupa* (Temminck), from South America (host reg. no. NMP P6V-100166), and there are no other data.

##### Type Material Deposition

The female holotype and most paratypes were deposited in the AMU (reg. no. AMU MS 22-1112-002), except two female paratypes that were deposited in the SNSB-ZSM.

##### Differential Diagnosis

This new species, collected from a host representative of the subfamily Tinaminae, differs from the other two described species, which were collected from host members of the subfamily Nothurinae, by the presence of short propodonotal setae *si* and hysteronotal setae *d2*. In females of *Tinamiphilopsis temmincki*, the setae *si* and *d2* lengths are 60–65 µm and 50–55 µm, respectively. In females of *Tinamiphilopsis elegans* Skoracki and Sikora, 2004, the lengths of setae *si* and *d2* are 160–205 µm and 150–185 µm, respectively, whereas in females of *Tinamiphilopsis ariconte* Skoracki et al., 2012, the setae *si* and *d2* are 155–165 µm and 110–125 µm long, respectively.

##### Etymology

The new species is named in honour of the Dutch ornithologist and naturalist Coenraad Jacob Temminck (1778–1858), who made significant contributions to the field of ornithology, particularly in the study and classification of bird species.

### 3.2. Parsimony Analysis

Three equally parsimonious trees were produced based on the initial analysis, with all characters being treated as unordered and unweighted (tree length 114, consistency index (CI) for phylogenetically informative characters—0.50, retention index (RI)—0.71, and rescaled consistency index (RC)—0.35); the character data and data matrix are presented in Appendix A. The strict consensus of these trees is shown in Figure 5. The differences between these trees lay only in the position of the genus *Trypetoptila* in relation to the genera *Crotophagisyringophilus*, *Syringophilopsis*, and *Torotrogla* (Figure 6). The successive weighting according to the rescaled consistency index did not change the topology of the strict consensus tree.

## 4. Discussion

To date, approximately 11,000 existing species are categorised as crown birds (Neornithes) [2]. These birds can be classified into two distinct and monophyletic groups: Palaeognathae (consisting of tinamous and ratites) and Neognathae (encompassing all other bird groups). Among the Neognathae, the Galloanserae (including Galliformes and Anseriformes) is considered the sister group to all other birds, referred to as the Neoaves [37,38,39]. Currently, syringophilid mites have been documented to inhabit 27 out of 44 orders of extant neognathous and paleognathous birds ([40], current study) (Figure 7).

### 4.1. Hypotheses on the Early History of Syringophilid Mites

The origin and the early evolution of birds and syringophilids associations is one of the most interesting aspects of the study of quill mites. It was hypothesised that syringophilid mites, which are similar to members of the family Cheyletidae, evolved from micro-predators that resided in bird nests or even the nests of theropod dinosaurs. Then, they migrated from such nests to feather quills [42,43]. Initially, the ancestors of syringophilids likely preyed upon other mites that inhabited wing vanes, like the modern cheyletid representatives of the tribe Cheletosomatini. It is worth noting that the majority of Cheletosomatini species are obligate predators residing in wing quills; however, mites from the genus *Picocheyletus* or *Metacheyletia* (the sole genus in the Metacheyletiini tribe) are likely parasites rather than predators in quills [44,45].

The “molecular clock” hypothesis suggests that the cheyletids and syringophilids diverged from each other approximately 180–185 million years ago, during the Early Jurassic period [46]. On the other hand, the earliest fossil widely accepted to belong to Neornithes, which includes all extant bird species, is *Vegavis* from the end-Cretaceous (~67 million years ago (Mya) [47]. However, numerous molecular dating studies have indicated that the diversification of Neornithes, which includes all extant bird species, likely started 100–110 million years ago [12,48,49]. In contrast, Prum et al. [8] presented findings, based on molecular clock analysis, that are congruent with the palaeontological record, supporting the major radiation of crown birds in the wake of the Cretaceous–Palaeogene (K–Pg) mass extinction (approximately 66 Mya). The facts mentioned above suggest that syringophilids likely had already formed relationships with the ancestors of birds, theropod dinosaurs, many of which had feathers, e.g., *Archaeopteryx* from the Late Jura [50,51,52] or *Aurornis* from the Middle-Late Jura [53].

### 4.2. Distribution of the Primitive Quill Mite Genera on the Host Lineages

The concept of coevolution was formally established as Fahrenholz’s rule by Eichler [54,55]. The simplest version of this rule is that “Parasite phylogeny mirrors host phylogeny” [56]. Coevolution is an appealing concept due to its simplicity and elegant explanatory power for the evolution of numerous parasites. Furthermore, in cases where coevolution takes place, the phylogeny of hosts can be inferred from the phylogeny of their parasites, and vice versa. This reciprocal relationship may offer valuable insights into the evolutionary dynamics of both hosts and parasites [57]. The expected similarities between host and parasite phylogenies, however, often do not exceed the similarity expected by chance between two random trees. This is because historical events (host switches, extinctions, etc.) often erode the expected patterns of co-speciation [58].

In 2004, Skoracki and Sikora [22] described the first species of syringophilid mites, *Tinamiphilus elegans*, collected from palaeognathous birds, the elegant crested tinamou. Eight years later, in 2012, Skoracki et al. [23] described the second species of this genus, *T. ariconte*, which was found on three tinamou hosts: the red-winged tinamou, the white-bellied nothura, and the lesser nothura. Taking into consideration that (i) syringophilids are obligate and permanent parasites; (ii) transmission occurs typically only when hosts come into direct physical contact, and most physical contact between individual hosts is between conspecifics, in particular between mates and between parents and offspring; (iii) many species of quill mite infect only a single or phylogenetically closely related species of host, and moreover, genera of syringophilids often are restricted to a single order of birds; (iv) representatives of the genus *Tinamiphilopsis* exhibit several primitive character conditions (e.g., smooth hypostomal apex, a large gnathosoma, edentate chelicerae, well-developed and sclerotised dorsal idiosomal shields, and full complement of idiosomal and leg setae); and (v) syringophilid mites exhibit high host specificity, the authors suggested that these discoveries support the hypothesis that the ancestor of the Syringophilidae transitioned to parasitism prior to the divergence of birds into the two major clades, Palaeognathae and Neognathae.

In 2013, Skoracki et al. [21] presented the first, but not fully resolved, phylogeny of syringophilid mites, where the genus *Tinamiphilopsis* was placed not as a sister lineage to the other syringophilid genera but in the core of the tree. These results contradicted the previous hypothesis [22,23] that the initial association of the genus *Tinamiphilopsis* was with Tinamiformes. The results obtained in the current study support the latter hypothesis. In the syringophilid tree, mites on the earliest derivate branches, i.e., *Selenonycha* Kethley and *Megasyringophilus* Fain et al., are associated with birds of the advanced clade Neoaves (Charadriiformes and Psittaciformes, respectively). In contrast, the mite genera associated with the earliest derivate clades of extant birds, Tinamiformes (Palaeognathae) and Galloanserae (Anseriformes and Galliformes), are mosaically distributed in the core of the tree (Figure 8). This contradiction between the presumable syringophilid parasitism of the common bird ancestor and the phylogenetic pattern obtained could be explained by the multiple switches (secondary infestation) from hosts of the Neoaves clade to palaeognathous and galloanserae birds and subsequent co-speciation.

## 5. Conclusions

In this research, we described a new species of syringophilid mite found in the feather quill of tinamou from the subfamily Tinaminae, the tataupa tinamou. This new species is easily distinguished from the other two species of the genus *Tinamiphilopsis* recorded from the representatives of the subfamily Nothurinae by the presence of the short setae *si* and *d2*. We also reconstructed the phylogeny of the most primitive genera of syringophilid mites, which showed incongruence with modern avian phylogenies. This suggests that host switching could play an important role in the early evolution of this group of mites. Furthermore, this study demonstrated that the mites of the genus *Tinamiphilopsis* originally parasitised Neoavian birds before moving to tinamous birds.

## Figures and Tables

**Figure 1 animals-13-02728-f001:**
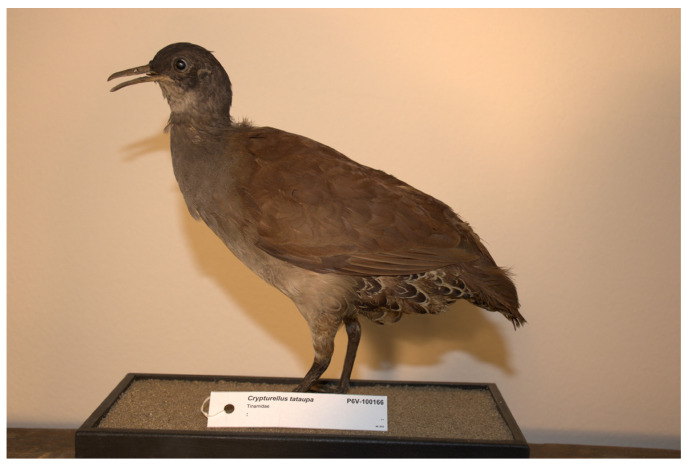
Host specimen of the tataupa tinamou *Crypturellus tataupa*, infested by *Tinamiphilopsis temmincki* sp. n.

**Figure 2 animals-13-02728-f002:**
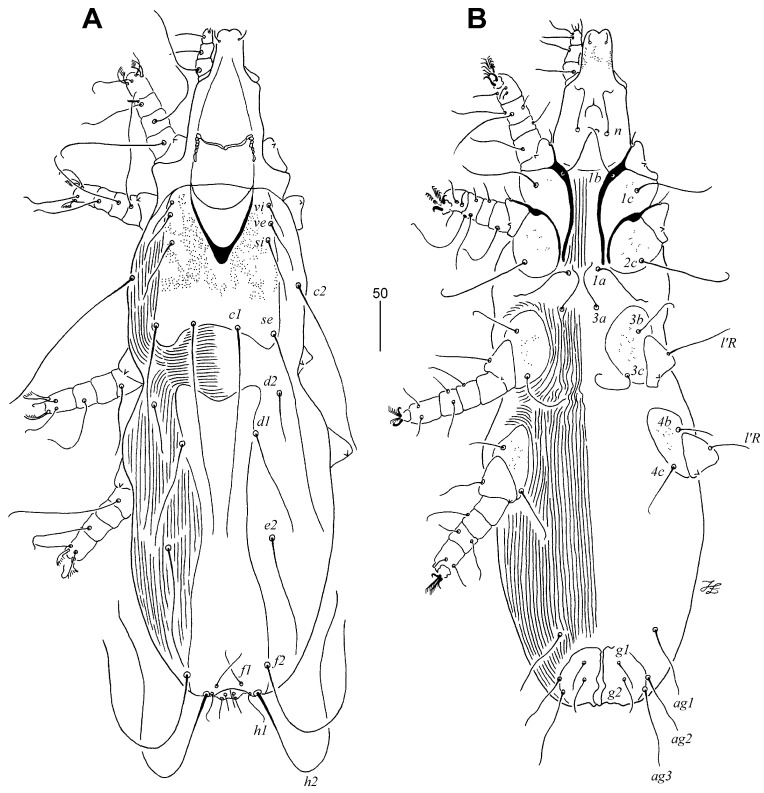
*Tinamiphilopsis temmincki* sp. n., female: (**A**) dorsal view and (**B**) ventral view.

**Figure 3 animals-13-02728-f003:**
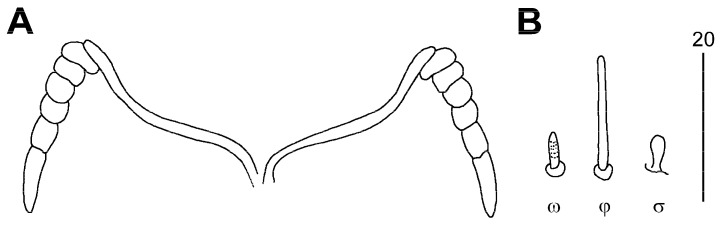
*Tinamiphilopsis temmincki* sp. n., female: (**A**) peritremes and (**B**) solenidia of leg I.

**Figure 4 animals-13-02728-f004:**
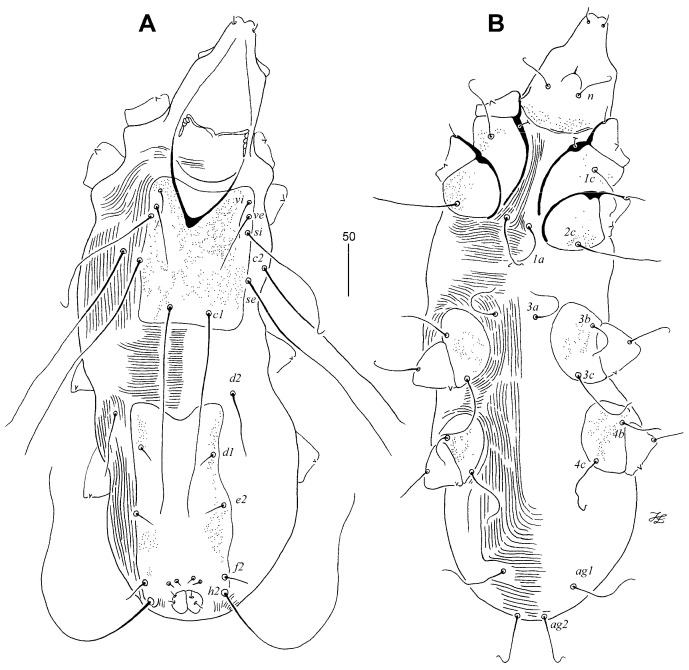
*Tinamiphilopsis temmincki* sp. n., male: (**A**) dorsal view and (**B**) ventral view.

**Figure 5 animals-13-02728-f005:**
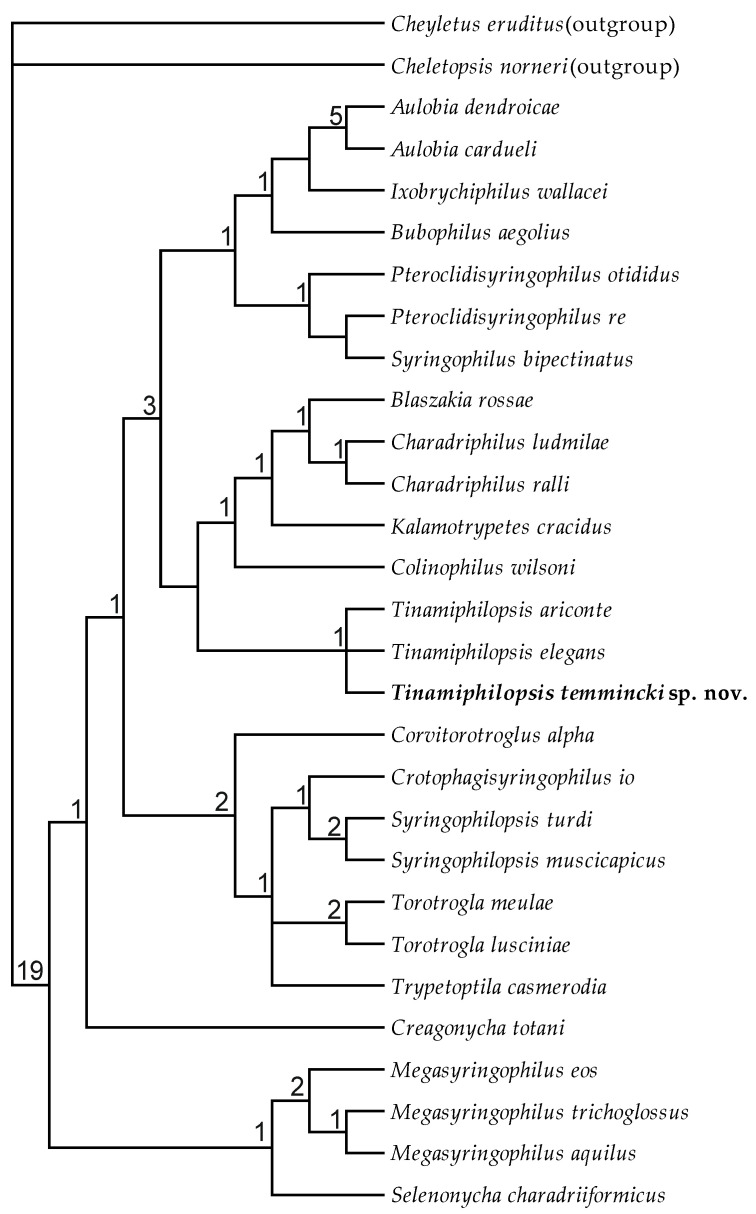
Strict consensus of the three most parsimonious trees (tree length 114, consistency index (CI) for phylogenetically informative characters—0.50, retention index (RI)—0.71, rescaled consistency index (RC)—0.35) found using the heuristic search option for the unordered and unweighted dataset. Numbers at nodes—Bremer indices.

**Figure 6 animals-13-02728-f006:**
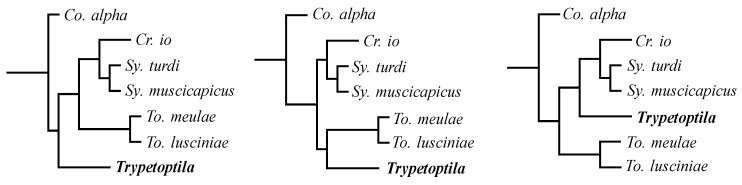
The differences between the topology of the three most parsimonious trees.

**Figure 7 animals-13-02728-f007:**
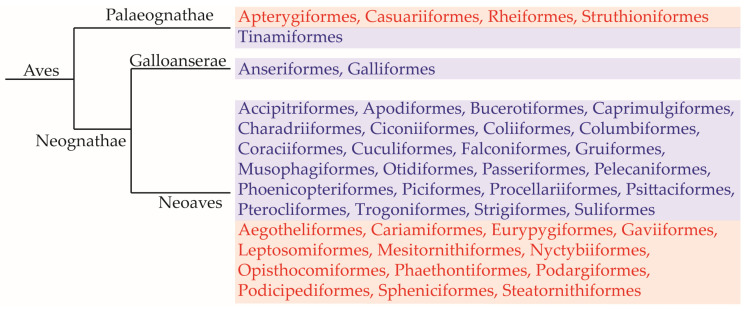
Phylogeny of birds with all extant orders (according to Sangster et al. [41]). Orders of birds on which syringophilid mites have been found are marked in blue; orders on which no syringophilids have been found thus far are marked in red.

**Figure 8 animals-13-02728-f008:**
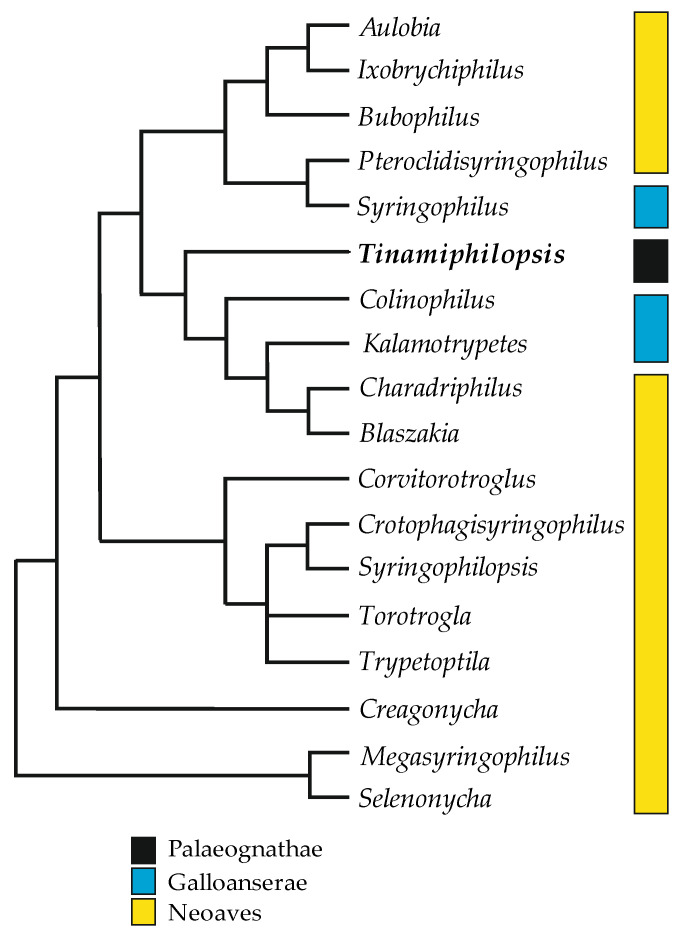
Distribution of the alpha mite genera on the main lineages of birds: Palaeognathae and Neognathae (Galloanserae + Neoaves).

## Data Availability

Data are available upon request from the corresponding author.

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
