# Peer review of "Tinamiphilopsis temmincki sp. n., a New Quill Mite Species from Tataupa Tinamou, and the Early History of Syringophilid Mites†"

_animals, 2023, doi:10.3390/ani13172728_

Round 1

Reviewer 1 Report

This paper has two parts. First, the description of a new species. Second, information on the origin and early history of Syringophilid mites. 

I feel the present title is too modest, it does not emphasize strongly the second part of the text. Therefore, I would recommend a title which is more straightforward (braver) and a bit shorter. Like this: 

"Tinamiphilopsis temmincki n.sp., a quill mite from the Tataupa Tinamou, and the early history of syringophilid mites" 

(Feel free not to accept my recommendation)

-----

lines 277-278 should also cite Fahrenhotz's original paper. 

Lines 282-283. In recent decades, a whole new set of methodologies have been developed to test this hypothesis. A large number of case studies indicate that co-speciation is rarely detectable by the comparison of host and parasite phylogenetic trees. 

Therefore, I recommend to add the word "may" to soften the phrase to "...reciprocal relationship may offer valuable..."

To the end of this paragraph, I would add 1-2 further sentences and a reference to mention these new results, something like this:

"The expected similarities between host and parasites phylogenies, however, often do not exceed the similarity expected by chance between two random trees. This is because historical events (host switches, extinctions etc.) often erode the expected patterns of co-speciation (Page RDM 2003. Tangled Trees ...).

It is worth to mention these new results, because they are in accordance with your present findings as well. 

Author Response

Dear Reviewer,

Thank you very much for the time dedicated and all the comments regarding our manuscript. We have taken all the suggestions into account and incorporated them into the text.

Point 1. I feel the present title is too modest, it does not emphasize strongly the second part of the text. Therefore, I would recommend a title which is more straightforward (braver) and a bit shorter. Like this: 

"Tinamiphilopsis temmincki n.sp., a quill mite from the Tataupa Tinamou, and the early history of syringophilid mites" 

(Feel free not to accept my recommendation)

>>>Answer. Thank you for this suggestion. We changed the title.

Point 2. lines 277-278 should also cite Fahrenhotz's original paper. 

>>>Answer. You're right; we added this paper to the references.

Point 3. Lines 282-283. In recent decades, a whole new set of methodologies have been developed to test this hypothesis. A large number of case studies indicate that co-speciation is rarely detectable by the comparison of host and parasite phylogenetic trees. 

Therefore, I recommend to add the word "may" to soften the phrase to "...reciprocal relationship may offer valuable..."

>>>Answer. You're right; we added "may" to the sentence.

Point 4. To the end of this paragraph, I would add 1-2 further sentences and a reference to mention these new results, something like this:

"The expected similarities between host and parasites phylogenies, however, often do not exceed the similarity expected by chance between two random trees. This is because historical events (host switches, extinctions etc.) often erode the expected patterns of co-speciation (Page RDM 2003. Tangled Trees ...).

It is worth to mention these new results, because they are in accordance with your present findings as well. 

>>>Answer. Thank you for the suggestion. We added the sentence to the text and citation of RDM Page to the references.

Best regards,

Authors

Reviewer 2 Report

This is a well-prepared description with a sound phylogenetic analysis. The paper is well-written, being clear and concise. I have made very minor corrections, made a minor note of interest, and thought that the last two sentences of the conclusion were a bit muddled. In essence, the paper is almost publishable as-is.

As noted above, the paper is clear and concise, so was easy to read and a pleasure to review.

Author Response

Dear Reviewer,

Thank you very much for the time dedicated and all the comments regarding our manuscript. We have taken all the suggestions into account and incorporated them into the text.

Point 1. This is a well-prepared description with a sound phylogenetic analysis. The paper is well-written, being clear and concise. I have made very minor corrections, made a minor note of interest, and thought that the last two sentences of the conclusion were a bit muddled. In essence, the paper is almost publishable as-is.

>>>Answer. Yes, you're right; we deleted these two sentences in the conclusion.

Best regards,

Authors